# Effects of the Eating Habits of Romanian Residents on the Water Footprint

Teodora Maria Rusu * , Tania Mihăiescu , Antonia Odagiu and Laura Paulette

Faculty of Agriculture, University of Agricultural Sciences and Veterinary Medicine Cluj-Napoca, Manastur Street 3-5, 400372 Cluj-Napoca, Romania; tania.mihaiescu@usamvcluj.ro (T.M.); antonia.odagiu@usamvcluj.ro (A.O.); laura.paulette@usamvcluj.ro (L.P.)

* Correspondence: teodora-maria.rusu@student.usamvcluj.ro; Tel.: +40-727-204-220

**Abstract:** Water footprint assessment is an analytical tool that helps us understand how activities, actions, and products from human activity influence the scarcity and pollution of water resources. The objectives of the paper are to study the water footprint that is necessary for the production of food for human consumption as an effective way to determine how food habits put pressure on water resources and to identify ways to reduce the stress found on them. To calculate the water footprint of food products consumed by Romanian residents, two types of data were used: information on the average annual net food consumption of each type of food considered during the research and the water footprint per unit of food consumed. In addition, an analysis was carried out based on the structure of the water footprint and the structure of food consumption. In terms of the structure of the water footprint, the contribution rate of the green water footprint is the highest, reaching 83.5%. This is followed by the blue water footprint and the gray water footprint, accounting for 9.04% and 7.46%, respectively. From the perspective of the structure of food consumption, the consumption of cereals, meat, milk, and dairy products contributed the most to the water footprint of residents' food consumption, reaching 21.8% and 26.6%, respectively, and contributing 24.2% to the total water footprint of food consumption. Our research is useful for water management, improving the efficiency of use in agricultural technologies, and optimizing the structure of food consumption, such as reducing grain and meat consumption.

**Keywords:** green water footprint; blue water footprint; gray water footprint; food consumption; eating habits





## 1. Introduction

Water plays an extremely important role in terms of the existence and support of life on Earth, also being a component of the global ecosystem [1]. This is a resource that not only meets the basic needs of the population but is also the key to development on various levels [2–5], supporting the economy through agriculture, commercial fishing, power generation, industry, shipping, and tourism. Although one might create the illusion of an abundance of this precious resource, less than 1% of all the resources are available as fresh water [6].

Due to the many pressures on water resources, such as those exerted by land use, water abstraction, and various industrial activities, they have undergone drastic changes [7–9]. Water scarcity is becoming an increasingly critical problem globally [10–13], playing a very important role in exacerbating climate change, health problems, and agricultural problems. Thus, solving the problem of water shortage is a complex challenge that requires innovative solutions and the involvement of both the public and private sector [14,15].

The management of water resources must start with an appropriate method of collecting data and quantifying the impact on them [16]. Therefore, the implementation of some technological, operational, and management concepts and tools for the evaluation

and for the decision support in the management of water resources is a necessary scientific approach in order to achieve the best results [17–19]. One such concept is the water footprint [20–23]. A water footprint is a geographically explicit indicator showing the consumption of fresh water by a population [21]. For agricultural products, it can be described as the volume of fresh water (blue, green, and gray) used to produce a product along its supply chain in each step of the production process. This contributes to an advanced degree of understanding of the impact that an activity has on water resources and constitutes a solid basis for making informed decisions and adopting integrated policies to reduce the impact on water [24].

The concept of a water footprint was proposed by Arjen Hoekstra in 2002 as a means of measuring the volumes of water consumed and polluted during the production of goods and services [25]. The water footprint comprises three main components [24,25]: the green water footprint (meteoric water that is stored in the soil and vegetation, evaporated, and transpired), the blue water footprint (fresh water that is consumed from surface or deep sources), and the gray water footprint (fresh water that is necessary for the assimilation of pollutants). Additionally, with the growth of international trade, the use of water resources has become disconnected from end consumers [25]. Therefore, the footprint of water includes both direct and indirect uses.

Tony Allan introduced the concept of "virtual water" in 1993 [26], which was used to quantify the amount of water consumed in the production of products or services. The concept was particularly relevant for the water management issues in the Middle East [27]. The term "virtual water" differs from what is known as physical water resources, and it is often known as embedded water or invisible water [27].

The concept of the water footprint that was later introduced by Professor Hoekstra is based on the concept of virtual water introduced by Tony Allan [28], because the water footprint in Hoekstra's sense includes both direct and indirect (virtual) water footprints. Thus, the water footprint of food consumption, for example, shows the amount of water resources that is necessary to produce the products, actions, and services consumed by a certain population in certain material standards of life and availability. The food we eat every day puts significant pressure on the available water resources [21,24]. This offers an actual, realistic representation of the actual amount of water resources used in food production, including the direct and indirect (virtual) water footprint [25,26]. Thus, in addition to raising awareness and integrating the problem of water scarcity into public concerns, access to and availability of data related to the volumes of water consumed and polluted are equally important, so that there are no delays and uncertainties due to insufficient data.

The purpose of this paper is to quantify how the dietary habits in Romania influence the water requirements. Taking into account the food habits identified for ordinary citizens in Romania, the research shows the level of pressure that they put on the water resources of the country, but also of other countries with which Romania trades. Fifteen categories of products are taken into account, which are identified from the food balance report of the Romanian National Institute of Statistics (RNIS) from 2021, of which eight are of vegetable origin and seven are of animal origin. Water scarcity is a global and growing problem. The production of agricultural goods requires a large amount of water, and the eating habits of consumers indirectly place an increasing pressure on water resources. Therefore, the study of the water footprint and its components required to produce food for human consumption is an effective way to determine how food habits put pressure on water resources and to identify ways, tools to reduce the pressure that is put on them.

The objectives of the paper are as follows: (1) to calculate the water footprint of food consumption by Romanian residents for the year 2020; (2) to perform an analysis of the water footprint based on the structure of the water footprint and the structure of food consumption; and (3) to identify ways to reduce the stress on water resources.

## 2. Materials and Methods

It is well known that citizens' food preferences significantly influence water demands [29]. This fact is due to the high volume of water needed to grow, cultivate, and process food that is part of an individual's daily intake.

To calculate and analyze the water footprint of the food products consumed by Romanian residents, two types of data were used:

(2.1.) information on the average annual net food consumption of each type of food considered during the research (expressed in kg inhabitant$^{-1}$) and

(2.2.) the water footprint per unit of food consumed.

### 2.1. Gross Average Annual Food Consumption of Each Type of Food

Depending on the food structure, their characteristics, and the availability of data on residents' food consumption, 15 categories of products were considered, of which 8 are of vegetable origin and 7 of animal origin, based on the data obtained from the food balance for the year 2020 of Romanian National Institute of Statistics (RNIS) from 2021. Thus, the gross average annual consumption of each type of food in a resident's usual diet can be found in Table 1.

**Table 1.** Gross average annual consumption based on food type.

| Type of Product | Average Gross Annual Food Consumption, kg Inhabitant$^{-1}$ |
|---|---|
| **Grains and derived foods** | 204.4 |
| 1. Wheat and rye | 160.5 |
| 2. Corn | 38.8 |
| 3. Other grains | 0.5 |
| 4. Rice | 4.6 |
| **Potatoes** | 93.4 |
| **Beans** | 3.6 |
| **Vegetables and derived foods** | 167.8 |
| 1. Tomatoes | 42.1 |
| 2. Onion | 20.7 |
| 3. Cabbage | 43.6 |
| 4. Root vegetables | 14.0 |
| 5. Other vegetables | 47.4 |
| **Melons** | 23.0 |
| **Fruit** | 107.6 |
| 1. Apples | 29.1 |
| 2. Plums | 7.9 |
| 3. Cherries and sound cherries | 4.1 |
| 4. Peaches and nectarines | 4.6 |
| 5. Grapes | 7.9 |
| 6. Other fruit | 14.5 |
| 7. Exotic fruit | 39.5 |
| **Sugar and derived foods** | 25.5 |
| **Milk and derived foods** | 252.6 |

**Table 1.** *Cont.*

| Type of Product | Average Gross Annual Food Consumption, kg Inhabitant$^{-1}$ |
|---|---|
| **Eggs** | 11.8 |
| **Meat and derived foods** | 74.1 |
| 1.    Bovine meat | 5.4 |
| 2.    Porcine meat | 37.3 |
| 3.    Ovine and caprine meat | 2.6 |
| 4.    Avian meat | 28.0 |
| 5.    Other types of meat | 0.8 |
| **Edible organs** | 3.3 |
| **Fish and derived foods** | 6.3 |
| **Vegetable oils** | 15.6 |
| **Porcine fats** | 2.4 |
| **Butter** | 1.5 |

*2.2. Water Footprint per Unit of Food Consumed*

Agricultural products, as well as quality and quantity, require different amounts and types of water depending on their type [12,16]. It may also happen that the same type of agricultural product requires different amounts of water resources depending on the region where it is produced. This is mainly due to regional differences in climatic conditions, biological characteristics, production technology, terrain topography, and soil characteristics [2].

The direct water footprint is the water used directly, day-to-day. The indirect water footprint is the water used in the production of the products we buy, the food we buy, the energy we use, or the gasoline we use to fuel a car. By considering both direct and indirect footprints as components of the total water footprint [24], a better and broader perspective is obtained on how a consumer or producer uses freshwater resources. The water footprint is a volumetric measure of water consumption and the pollution it creates in this process [21]. This is not necessarily a measure of the severity of the impact on local environmental resources of water consumption. The local environmental impact of a given amount of water consumption and pollution depends on the vulnerability of the local water system and the number of consumers and polluters using the same system. Thus, the water footprint provides explicit information in time and space about how water is used for different purposes [8,11].

Currently, the most detailed and comprehensive research on the water footprint of agricultural products can be found in reports no. 47 and 48 published by Hoekstra and Mekonnen, who studied the water footprint of different agricultural products from different countries and administrative regions around the world [30,31]. The data relevant to the calculation methodology found in reports 47 and 48 (Table 2) were used in this paper.

According to Table 2, the average water footprint for cereal crops is 1644 m$^3$ ton$^{-1}$. Among the cereals studied, the footprint for wheat is relatively large (1827 m$^3$ ton$^{-1}$) due to the long vegetation period, while for corn it is relatively small (1222 m$^3$ ton$^{-1}$) compared to wheat. The average water footprint of rice, one of the world's most important cereals, is similar to the average water footprint of all cereal crops [32].

**Table 2.** Decomposed and total water footprint of food products present in Romania's food balance for 2020.

| Product Type | Water Footprint, m³ ton⁻¹ | | | |
|---|---|---|---|---|
| | **Green** | **Blue** | **Gray** | **Total** |
| **Grains and derived foods** | | | | |
| 1. Wheat and rye | 1277 | 342 | 207 | **1827** |
| 2. Corn | 947 | 81 | 194 | **1222** |
| 3. Other grains | 1645 | 38 | 113 | **1795** |
| 4. Rice | 1146 | 341 | 187 | **1673** |
| **Potatoes** | 191 | 33 | 63 | **287** |
| **Beans** | 3945 | 125 | 983 | **5053** |
| **Vegetables and derived foods** | | | | |
| 1. Tomatoes | 108 | 63 | 43 | **214** |
| 2. Onion | 192 | 88 | 65 | **345** |
| 3. Cabbage | 181 | 26 | 73 | **280** |
| 4. Root vegetables | 106 | 28 | 61 | **195** |
| 5. Other vegetables | 195 | 27 | 104 | **326** |
| **Melons** | 147 | 25 | 63 | **235** |
| **Fruit** | | | | |
| 1. Apples | 561 | 133 | 127 | **822** |
| 2. Plums | 1570 | 188 | 422 | **2180** |
| 3. Cherries and sound cherries | 961 | 531 | 112 | **1604** |
| 4. Peaches and nectarines | 583 | 188 | 139 | **910** |
| 5. Grapes | 425 | 97 | 87 | **609** |
| 6. Other fruit | 370 | 148 | 114 | **632** |
| 7. Exotic fruit | 660 | 97 | 33 | **790** |
| **Processed sugar** | 1184 | 487 | 111 | **1782** |
| **Milk and derived foods** | 1425 | 51 | 57 | **1533** |
| **Eggs** | 2579 | 106 | 107 | **2792** |
| **Meat and derived foods** | | | | |
| 1. Bovine meat | 17,938 | 541 | 614 | **19,093** |
| 2. Porcine meat | 4443 | 276 | 227 | **4946** |
| 3. Ovine and caprine meat | 8113 | 526 | 162 | **8801** |
| 4. Avian meat | 3666 | 166 | 147 | **3979** |
| 5. Other types of meat | 4299 | 272 | 219 | **4790** |
| **Edible organs** | 476 | 28 | 24 | **528** |
| **Fish and derived foods** | 1638 | 178 | 158 | **1974** |
| **Vegetable oils** | 7182 | 667 | 366 | **8215** |
| **Porcine fats** | 3666 | 166 | 147 | **3979** |
| **Butter** | 6671 | 239 | 269 | **7179** |

*2.3. Calculation of the Water Footprint*

The per capita water requirement (footprint) for food represents the amount of water used to produce the food required for one resident's consumption [33]. The water footprint

is calculated by multiplying the average annual consumption of each type of food (found in Table 1) by the virtual water footprint of the corresponding food (found in Table 2). The final result is obtained by summing the results obtained for all the food categories studied.

For example, the total water footprint of potato consumption per capita in Romania is obtained as follows:

$$\text{Water footprint }_{\text{potatoes}} = \text{average annual potato consumption} \times \text{total water footprint }_{\text{potatoes}} \tag{1}$$

$$\text{Water footprint }_{\text{potatoes}} = 0.0934 \text{ ton inhabitant }^{-1} \text{ year }^{-1} \times 287 \text{ m}^3 \text{ ton }^{-1} \tag{2}$$

$$\text{Water footprint }_{\text{potatoes}} = 26.81 \text{ m}^3 \text{ capita }^{-1} \text{ year }^{-1} \tag{3}$$

The evaluation of the water footprint includes four important stages: i. specifying the objectives and the main purpose of the evaluation; ii. water footprint accounting (quantification of the water footprint in space and time); iii. evaluating the sustainability of the water footprint (from a social, economic, and ecological point of view); and iv. formulating answers and conclusions.

## 3. Results

The results of the water footprint values for each type of food, as well as the total water footprint of food consumption per capita in Romania, can be found in Table 3.

**Table 3.** Results water footprint per capita.

| Product Type | Water Footprint Per Capita, m³ Inhabitant⁻¹ Year⁻¹ | | | |
| --- | --- | --- | --- | --- |
| | **Green** | **Blue** | **Gray** | **Total** |
| **Grains and derived foods** | 247.79 | 59.62 | 41.67 | **349.24** |
| 1. Wheat and rye | 204.96 | 54.89 | 33.22 | **293.23** |
| 2. Corn | 36.74 | 3.14 | 753 | **47.41** |
| 3. Other grains | 0.82 | 0.02 | 0.06 | **0.9** |
| 4. Rice | 5.27 | 1.57 | 0.86 | **7.7** |
| **Potatoes** | 17.84 | 3.08 | 5.88 | **26.81** |
| **Beans** | 14.20 | 0.45 | 3.54 | **18.19** |
| **Vegetables and derived foods** | 27.13 | 7.27 | 12.12 | **46.52** |
| 1. Tomatoes | 4.55 | 2.65 | 1.81 | **9.01** |
| 2. Onion | 3.97 | 1.82 | 1.35 | **7.14** |
| 3. Cabbage | 7.89 | 1.13 | 3.18 | **12.21** |
| 4. Root vegetables | 1.48 | 0.39 | 0.85 | **2.73** |
| 5. Other vegetables | 9.24 | 1.28 | 4.93 | **15.45** |
| **Melons** | 3.38 | 0.57 | 1.45 | **5.41** |
| **Fruit** | 70.15 | 15.15 | 11.13 | **96.43** |
| 1. Apples | 16.33 | 3.87 | 3.7 | **23.92** |
| 2. Plums | 12.4 | 1.49 | 3.33 | **17.22** |
| 3. Cherries and sound cherries | 3.94 | 2.18 | 0.46 | **6.58** |
| 4. Peaches and nectarines | 2.68 | 0.86 | 0.64 | **4.19** |
| 5. Grapes | 3.36 | 0.77 | 0.69 | **4.81** |
| 6. Other fruit | 5.37 | 2.15 | 1.65 | **9.16** |
| 7. Exotic fruit | 26.07 | 3.83 | 1.3 | **31.21** |

**Table 3.** *Cont.*

| Product Type | Water Footprint Per Capita, m³ Inhabitant⁻¹ Year⁻¹ | | | |
| --- | --- | --- | --- | --- |
| | Green | Blue | Gray | Total |
| **Processed sugar** | 30.19 | 12.42 | 2.83 | **45.44** |
| **Milk and derived foods** | 359.95 | 12.88 | 14.4 | **387.24** |
| **Eggs** | 30.43 | 1.25 | 1.26 | **32.95** |
| **Meat and derived foods** | 389.77 | 19.45 | 16.51 | **425.73** |
| 1.　Bovine meat | 96.87 | 2.92 | 3.32 | **103.10** |
| 2.　Porcine meat | 165.72 | 10.29 | 8.47 | **184.49** |
| 3.　Ovine and caprine meat | 21.09 | 1.37 | 0.42 | **22.88** |
| 4.　Avian meat | 102.65 | 4.65 | 4.12 | **111.41** |
| 5.　Other types of meat | 3.44 | 0.22 | 0.18 | **3.83** |
| **Edible organs** | 1.57 | 0.09 | 0.08 | **1.74** |
| **Fish and derived foods** | 10.32 | 1.12 | 0.99 | **12.44** |
| **Vegetable oils** | 112.04 | 10.41 | 5.71 | **128.15** |
| **Porcine fats** | 8.8 | 0.4 | 0.35 | **9.55** |
| **Butter** | 10.01 | 0.36 | 0.4 | **10.77** |
| **Food Consumption** | 1333.57 (83.50%) | 144.52 (9.04%) | 119.18 (7.46%) | **1597.27 (100%)** |

Compared to the average water consumption per capita in other countries, the water footprint of Romanian residents' food consumption is relatively small: 1597.27 m³ year⁻¹ capita⁻¹ [34]. Industrialized countries have water footprints in the range of 1250–2850 m³ year⁻¹ capita⁻¹, while it is necessary to optimize developing countries because they show a much larger range of 550–3800 m³ year⁻¹ capita⁻¹ [35].

The broken down calculated percentage composition of the total water footprint of food consumption per capita in Romania is presented in Figure 1. The water footprint of food consumption by the residents sums up the green water, blue water, and gray water footprints. Thus, 83.5% of the water consumed in food production is represented by green water. Even in irrigated agriculture, green water often makes a very significant contribution to total water use [36]. The large fraction of green water required to produce food for consumption confirms its importance in global food production. The blue water fraction is lower (9.04%), followed by the gray water fraction (7.46%).

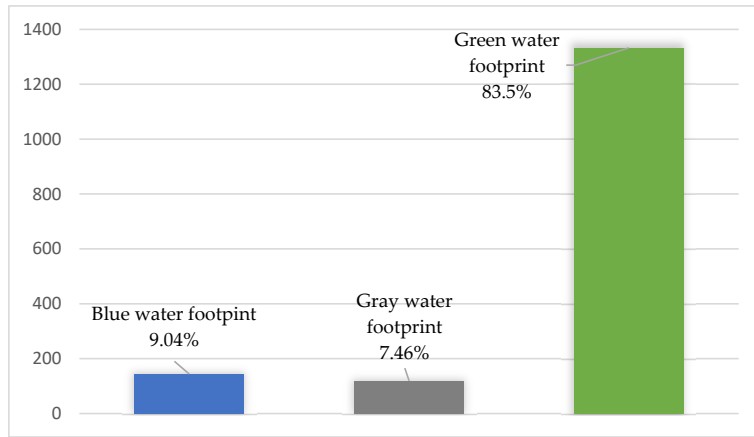

**Figure 1.** The structure of the food consumption water footprint of Romanian residents (m³ inhabitant⁻¹ year⁻¹).

The water footprint of residents' food consumption by type is shown in Figure 2. From the perspective of the water footprint structure of food consumption, the consumption of cereals, meat, and milk products contributed the most to the water footprint of residents' food consumption, contributing 21.8% and 26.6%, respectively, and 24.2% to the total water footprint of food consumption. After these, refined vegetable oils and fruits followed.

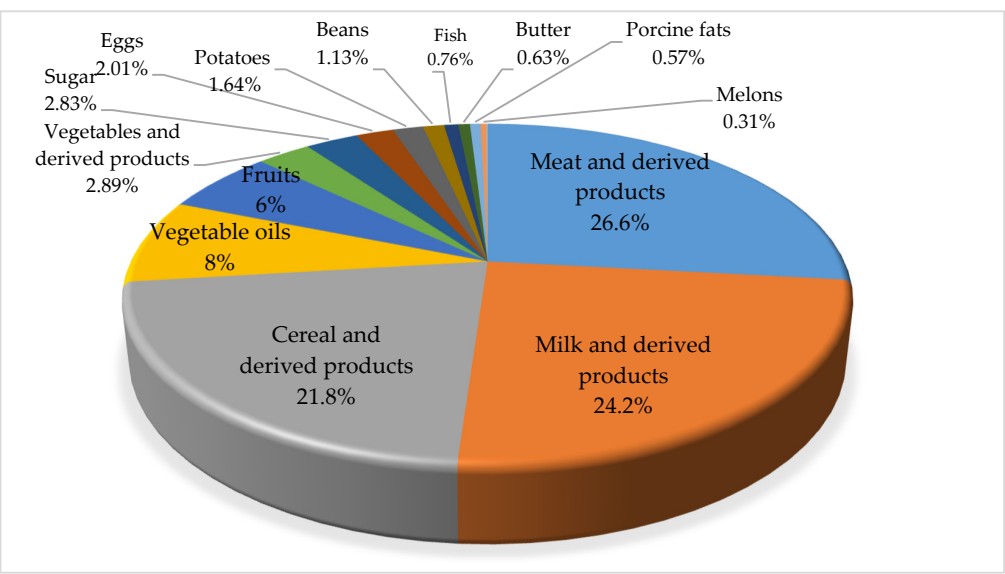

**Figure 2.** The proportion of the water footprint consumed by each type of food in Romania (%).

The water footprint of residents' meat consumption by type is shown in Figure 3. Thus, among the types of meat consumed by Romanian residents, pork contributed the most to the total water footprint of meat consumption, representing 44.5% of the total. This was followed by avian meat with 26.1% and bovine meat with 24.2%.

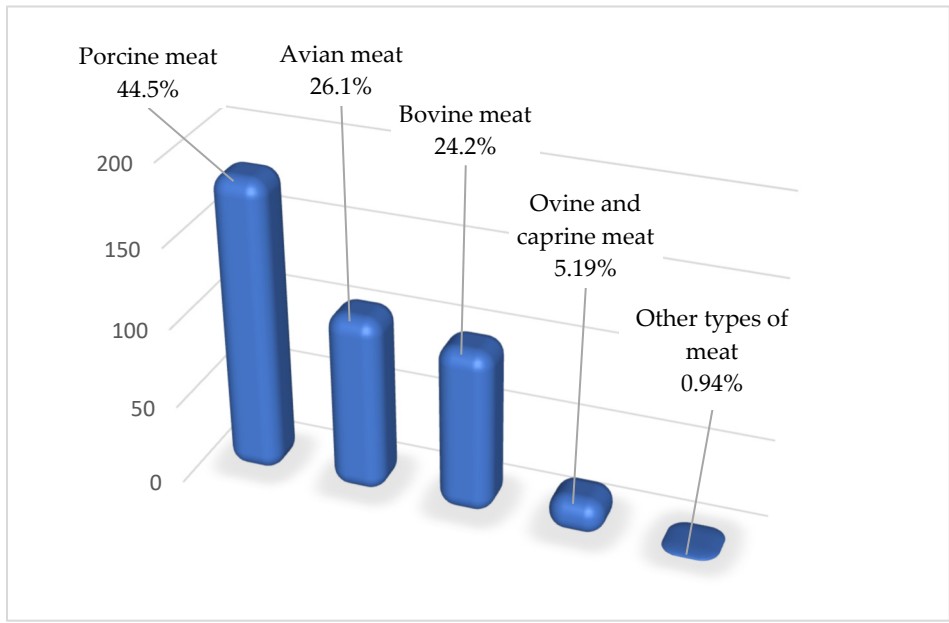

**Figure 3.** The water footprint of each type of meat in Romania residents (m$^3$ inhabitant$^{-1}$ year$^{-1}$).

## 4. Discussion

The study of the water footprint that is necessary for the production of food for human consumption is an effective way to determine how food habits put pressure on water resources and to identify ways to reduce/alleviate the stress found on them [28,37,38]. Hoekstra and Chapagain [39] showed that determining the indirect use of water required to produce goods can help understand the global nature of freshwater resources and quantify the effects of consumption and trade on water resource use. Such advanced understanding due to this newly introduced concept of water footprint can form the basis for better management of the globe's freshwater resources.

The green water footprint refers to the water from precipitation consumed in the production of an agricultural product, the blue water footprint refers to the surface and underground water consumed (evaporated), and the gray water footprint shows the volume of polluted water, i.e., the volume of fresh water that is necessary to assimilate and neutralize pollutants based on existing water quality standards [35,40–42]. As an indicator of "water use", the water footprint differs from the classical method of measuring "water uptake" in three ways: (1) It does not include the use of blue water to the extent that this water is returned from where it came; (2) it is not limited to the use of blue water, but also includes green and gray water; and (3) it is not limited to direct water use but also includes indirect water use.

The green water footprint represents the volume of water resulting from precipitation, that accumulates in the plant root zone and is ultimately consumed by the plant, which takes it up from the soil solution and biological growth, respectively, through the process of evapotranspiration [35]. In short, it represents the volume of water from precipitation consumed for biological processes during the plant production process. In the case of our study, this represents 1333.57 m$^3$ inhabitant$^{-1}$ year$^{-1}$, which represents 83.50%, so with the largest share. By category, the highest values are recorded for meat and derived foods (389.77 m$^3$ inhabitant$^{-1}$ year$^{-1}$), milk and derived foods (359.95 m$^3$ inhabitant$^{-1}$ year$^{-1}$), and grains and derived foods (247.79 m$^3$ inhabitant$^{-1}$ year$^{-1}$).

The blue water footprint represents the amount of water that has been used from the surface or groundwater, such as lakes, rivers, and aquifers, and in our case, has been incorporated into an agricultural product [39,43]. The blue water footprint can refer to irrigated agriculture, water use in secondary processes, domestic water use, and water use in commercial processes. In the case of our study, this represented 144.52 m$^3$ inhabitant$^{-1}$ year$^{-1}$, representing a share of 9.04%. It should be noted that in this case, the highest share is recorded in rains and derived foods, with 59.62 m$^3$ inhabitant$^{-1}$ year$^{-1}$.

The gray water footprint refers to the amount of water used to dilute and neutralize pollutants, such as secondary processes in agricultural production, agricultural waste, industrial waste, and urban waste, in order to reach and fulfill the quality standards required for the products obtained [30–32]. The gray water footprint of a final product is an indicator of the level of freshwater pollution that can be attributed to and associated with the production of an agricultural product (in our case) along its entire chain of preparation, supply, and production [43]. It is defined as the volume of water required to dilute and neutralize the pollutants resulting from the production process to such an extent that the water quality, after this process, meets the conventionally established quality standards. In the case of Romania, this represents 119.18 m$^3$ inhabitant$^{-1}$ year$^{-1}$, representing a share of 7.46%. The highest value was recorded for grains and derived foods with 41.67 m$^3$ inhabitant$^{-1}$ year$^{-1}$.

Currently, Romania's water resources, which are sufficient in terms of quantity, face various types of difficulties in terms of distribution, appropriate use, time, and space, as well as the management of groundwater, surface water, intermediate water, and new water, as well as water competition for agriculture, industry, and tertiary industry [44]. In general, the contradiction between the demand and supply of water resources is very prominent; its use is not optimized, and water pollution is serious. With agriculture being one of the occupations that is the basis of the Romanian economy, it is a big consumer of

water resources. The water footprint is the connection point that links water resources, agricultural products, and agricultural policies in the future [2,30]. Thus, since the study shows that grains and derived foods register the highest values for blue and gray water, it shows the need to rethink the continuation of agricultural policies in this direction, as is currently happening.

Following the research that was carried out, we can deduce the fact that, from the perspective of the structure of food consumption, in order to reduce the total food consumption footprint of the residents, the key is to reduce the consumption of grains and meat.

Among meat products, beef has the largest water footprint, requiring 15,400–19,093 m$^3$ of water to produce one ton of meat [24,43]. The water footprint for wheat is relatively large, while for corn it is relatively small, and in the case of fruits, we find a similar variation. Additionally, refined vegetable oils are heavy consumers of water resources, requiring 8280 m$^3$ of water to produce one ton of oil [45–47]. Agricultural products require different amounts of water depending on their type [22,48,49]. It can also be the case that the same type of agricultural product requires different amounts of water resources depending on the region where it is produced, mainly due to regional differences in climatic conditions, production technology, and soil characteristics [50–52].

In order to reduce the water footprint related to food consumption, it is necessary to promote and develop the concept of sustainable consumption. Sustainable consumption aims at various levels of action and involves approaches concerned with the development, implementation, and popularization of consumption practices and production innovations intended to reduce the environmental impact and negative social effects of economic activities [48]. Sustainable consumption supports education and information activities regarding environmental costs and the dependence links between resources, production, and consumption, as well as the hidden costs of the consumerist lifestyle [53]. In addition, it is necessary to control the consumption of large water-consuming products consciously and correctly [53–55]. On the other hand, a big problem and a means of intervention related to food consumption is food waste [56–58]. A large amount of food is lost or wasted in both the production and consumption processes [59–61]. Therefore, to reduce our water footprint, we must raise awareness and take appropriate measures to reduce food waste [54,62]. By highlighting the importance of accounting for the water required for food consumption, the foundations for the education necessary to adopt sustainable consumption habits can be laid [63,64].

The final utility of this study is very diverse [65–67]: raising awareness and sounding the alarm for citizens, entrepreneurs, and politicians; identifying activities that consume the water resource excessively; formulating policies to reduce water scarcity; and many other uses. Analyzing water use through a methodology applied only in their own country, governments do not have a comprehensive vision and results on understanding the sustainability of national consumption [68,69]. Calculating a product's virtual water footprint also takes into account the supply chains and trade flows that underpin today's imports and exports [70]. This is a limitation of the present study and reveals the need for further research. This is because knowledge of water resource dependence elsewhere is relevant to the knowledge of a national government not only when assessing its national environmental policy [56] but also when assessing national or regional food security [13,71].

## 5. Conclusions

The study is useful for the development of scientific knowledge and efficient water management in Romania, but also in other countries, for highlighting and raising awareness of the need to improve agricultural technology, differentiated by area, in order to reduce the gray water footprint. At the same time, this study is also useful for raising awareness of the need to optimize and promote the food consumption structure of citizens, such as by reducing grain and meat consumption. Thus, the evaluation of the water footprint plays the role of a useful and significant tool for optimizing the sustainable management of water resources used in agriculture and ultimately in food production systems.

The main specific factors underlying the knowledge of the water footprint of national consumption are as follows: (1) the volume of water and the pattern of national consumption, and (2) the resulting water footprint per ton of products consumed. The second factor, in the case of agricultural products, depends on the specific topography, soil, climate, and applied technology, such as irrigation and fertilization practices and crop yields. The national average of the water footprint related to the consumption of food products in Romania for the year 2020 is 1597.27 m$^3$ year$^{-1}$ per capita. Regarding the structure of the water footprint, it is important to mention that the contribution rate of the green water footprint is the highest, reaching 83.5%. This is followed, with lower values, by the blue water footprint and then the gray water footprint, representing 9.04% and 7.46%, respectively. From the perspective of the structure of food consumption, specific to Romania, the consumption of cereals, meat, milk, and dairy products contributed the most, as the study shows, to the water footprint of the food consumption of the inhabitants, contributing 21.8% and 26.6%, respectively, and 24.2% to the total water footprint of food consumption.

The limitations of this study come from the choice of methods used in the analysis. Even though the comparative descriptive statistical analysis was indeed useful in terms of summarizing the data and identifying patterns and trends in Romania's food consumption, it does not provide any information on causality (why certain trends and patterns exist in the data). Thus, the research must be continued in order to identify and parameterize the intrinsic factors in production and consumption in Romania that can lead to sustainable consumption through the use of other statistical methods for data analysis.

**Author Contributions:** Conceptualization, T.M.R. and T.M.; methodology, T.M.R.; validation, T.M. and A.O.; resources, T.M. and L.P.; data curation, A.O.; writing—original draft preparation, T.M.R.; writing—review and editing, T.M.R.; visualization, T.M.; supervision, A.O. and L.P.; project administration, T.M. and L.P. All authors have read and agreed to the published version of the manuscript.

**Funding:** This research is financially supported by the University of Agricultural Sciences and Veterinary Medicine Cluj-Napoca through the Doctoral School Program.

**Data Availability Statement:** The data presented in this study are available on request from the corresponding author.

**Acknowledgments:** The authors would like to thank the University of Agricultural Sciences and Veterinary Medicine Cluj-Napoca for support and cooperation.

**Conflicts of Interest:** The authors declare no conflict of interest.

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
