# Peer review of "Effects of the Eating Habits of Romanian Residents on the Water Footprint"

_water, doi:10.3390/w15081622_

Round 1

Reviewer 1 Report

I have liked reading the paper and I think it's worth of pubblication. Just few things are to be changed, as well as some minor mistakes regarding English. Only the introduction chapter needs to be implemented, the others are fine to me.

There was no numbering for the lines and this does not help the review. Please fix it for the next time.

Please find the file attached for further comments.

Reviewer 2 Report

The article addresses a topic that is currently relevant but not new, the water footprint in food. It is a study without field work based on secondary data obtained mainly from two previous studies.

The abstract and keywords are adequate and reflect the subsequent content of the article.

The introduction to the topic of the water footprint is acceptable without delving into technical components that explain the concept

The methodology used is simple but correctly used, a basic comparative descriptive statistical analysis.

The results obtained lack an academic foundation, but it is due to the type of methodology used and the nature of the calculations.

The discussion is well justified and the conclusions based on the results obtained and on the subsequent discussion.

In the conclusions section, I think it is necessary to include a section with the limitations that this type of research poses.
